# Alternative Process Routes to Manufacture Porous Ceramics—Opportunities and Challenges

**DOI:** 10.3390/ma12040663

**Published:** 2019-02-22

**Authors:** Uwe Scheithauer, Florian Kerber, Alexander Füssel, Stefan Holtzhausen, Wieland Beckert, Eric Schwarzer, Steven Weingarten, Alexander Michaelis

**Affiliations:** 1Fraunhofer Institute for Ceramic Technologies and Systems IKTS, 01277 Dresden, Germany; florian.kerber@gmx.de (F.K.); alexander.fuessel@ikts.fraunhofer.de (A.F.); wieland.beckert@ikts.fraunhofer.de (W.B.); eric.schwarzer@ikts.fraunhofer.de (E.S.); steven.weingarten@ikts.fraunhofer.de (S.W.); alexander.michaelis@ikts.fraunhofer.de (A.M.); 2Chair of Engineering Design and CAD, Technical University Dresden, 01069 Dresden, Germany; stefan.holtzhausen@tu-dresden.de

**Keywords:** porous, ceramics, additive manufacturing, multimaterial, multiproperty, CerAMfacturing, CerAM VPP, CerAM T3DP, CerAM Replica

## Abstract

Porous ceramics can be realized by different methods and are used for various applications such as cross-flow membranes or wall-flow filters, porous burners, solar receivers, structural design elements, or catalytic supports. Within this paper, three different alternative process routes are presented, which can be used to manufacture porous ceramic components with different properties or even graded porosity. The first process route is based on additive manufacturing (AM) of macro porous ceramic components. The second route is based on AM of a polymeric template, which is used to realize porous ceramic components via replica technique. The third process route is based on an AM technology, which allows the manufacturing of multimaterial or multiproperty ceramic components, like components with dense and porous volumes in one complex-shaped component.

## 1. Introduction

Porous ceramics can be classified in terms of their porosity: mainly closed or open. In the case of closed porosity, the single pores are not or are only partially connected to each other and are not accessible from the outer surface. This kind of porous material can be advantageous for lightweight components or applications where a high thermal shock resistance is necessary. In contrast, ceramics with open porosity can be classified as a network of syndetic pores or cells. They are interesting for applications where a defined permeability for various fluids is required. For both types, open and closed pores, the pore’s size, size distribution, volume, and shape determine the properties of the porous ceramics.

Porous ceramics can be provided by different methods. One way is the sintering of ceramic powders without significant densification or the utilization of solid-state reactions of powder mixtures that lead to pore formation [1]. The resulting free space between the ceramic particles leads to a porosity from a few nanometers up to some micrometers (sometimes even millimeters), depending on the particle size distribution. Using this method, a wide range of pore volumes up to 75% is achievable [2]

Structures with even higher porosity are often called foams, which can be prepared by burning out pore formers, the direct foaming or the replication of a sacrificial template [3]. For the burnout of pore formers, natural organic, or synthetic polymer particles are applied, that evaporate or burn during the sintering step of the ceramic material. This technique leads to pores of some micrometers up to some millimeters [4,5]. Porous ceramics, prepared by direct foaming via physical or chemical foaming mechanisms reach pore volumes up to 45–85% [6]. However, due to destabilizing mechanisms such as coalescence and drainage, the pore size of those foams is often inhomogeneous distributed, which includes a non-negligible amount of closed pores [7].

An open pore volume of up to 95% can be achieved by the replica technique (Schwartzwalder method) [8]. Due to its high process stability, the possibility of automated and large production capacity, it is the most common process to fabricate open-celled foams. The result of the replication technique is an extremely open network of interconnected, but hollow struts. Depending on the chosen template, the structure can be regular or irregular.

Open-celled ceramic foams prepared by replication start from reticulated, netlike polymer foams, which are commercially available in various cell sizes. Classified by pores-per-inch (ppi) according to the ASTM standard D3576-77, a wide range of polymeric foams is applicable: coarse ones with about 5 ppi have a medium pore size of around 5 mm while the pores of fine foams with down to 90 ppi reach only ~200 µm.

The unique structure of open-celled foams leads to outstanding properties like high specific surface area, low pressure drop, low density, and high thermal shock stability. On that account, ceramic foams can not only be used for metal melt filtration, which is state-of-the-art in the casting industry since decades [9], but also for a large number of highly sophisticated applications. Nowadays, these are manly porous burners [10], solar receivers [11], structural design elements, or catalytic supports [12]. In all those applications, an adjusted flow behavior as well as a high mechanical and abrasion stability is mandatory. In addition, a high thermal and chemical resistance can be essential to reach increasing application temperatures in aggressive chemical environments.

Based on a statistically grown pore structure, ceramic foams are limited in terms of cell size, strut thickness and surface area determined by the polymeric foam templates. Alternatively, technical textiles and specifically additive manufactured templates enlarge the range of possible applications. Latter is one topic of this manuscript and will be described in detail below.

According to Scopus more than 16,000 publications about porous ceramics have been registered in the last 20 years and thereby more than one third in the last five years. This underlines the huge interest in this topic, driven by industrial applications and the actuality of the research. Based on an application based review, Hammel et al. [13] provided a good overview about different advantageous fields of application for porous ceramics from thermal insulation to filtration purposes and tissue engineering. Within this paper different alternative process routes are presented, which can be used to manufacture porous ceramic components with different properties. The first process route is based on additive manufacturing (AM) of porous ceramic components. The main challenge is to generate the necessary CAD (computer-aided design) data. Two different strategies are presented to create CAD data for ceramic components with very complex geometries. The second process route is based on AM of a polymeric template, which is used to manufacture porous ceramic components via replica technique. The third process route is based on an AM technology, which allows the manufacturing of multimaterial or multiproperty ceramic components, like components with dense and porous volumes in one complex-shaped component.

To highlight the AM of ceramic components towards the common and well-known AM techniques for polymers, we add the acronym “CerAM” to the names of the different techniques. “CerAM” stands for “CerAMfacturing” which summarizes all AM techniques which are used for the additive manufacturing of ceramic components at Fraunhofer Institute for Ceramic Technologies and Systems (IKTS) Dresden [14].

## 2. Alternative Process Routes

### 2.1. CerAMfacturing of Single-Material Porous Ceramics

For the first process route the CerAM VPP technology (vat photopolymerization) and a commercially available AM device (CeraFab 7500, Lithoz, Vienna, Austria) of Lithoz, Vienna is used. A highly particle-filled photocurable suspension is placed on the entire surface of a tub and selectively exposed to light. As a result, the current layer cures and connects with the layer which was manufactured before [15]. After the component has been completely built up by AM, it has to be debinded and sintered to realize the final ceramic component. Suspensions are commercially available for different ceramics, like alumina or zirconia, which allow the manufacturing of nearly dense ceramic components [16]. The final components may have very complex geometries [17,18] and convince by very smooth surfaces and highly accurate features.

One strength of additive manufacturing is that it allows completely new structures, which are not restricted by the principles of traditional design. But the workflow of most current CAD tools is focused on such traditional rules and concepts and so the setup of geometry models of some innovative structures for additive manufacturing is sometimes rather limited and laborious. A promising approach for efficiently creating structures—in particular, periodic and/or graded ones—with versatile shapes and features may be based on processing parametric surfaces from mathematical field functions. Isosurfaces may be used to simply create manifold structure surfaces and to separate domains. A huge variety of suitable mathematical functions exists (gyroids, lidinoids, harmonic surfaces, etc.); the variety of possible structural shapes may be further expanded by function combination and composition. Deliberate choice and variation of functional parameters allow us to adjust and modify the structures to desired specification targets (volume content, cell size, wall thickness, etc.), and even the creation of graded structures often turns out to be a simple operation by spatial variation of parameters. The same principles (isosurfaces) in structure creation may also be applied to field functions which result from a numeric analysis of mathematical-physical models, representing a link to automated and straight forward structural design from model based component optimization techniques (topology optimization). Traditional, NURB kernel-based CAD tools have only limited capabilities to directly generate such structures and the process of geometry import from external software tools using stereolithographic formats (*.stl) may be very labor-intensive. 

An alternative CAD approach that is perfectly compatible to the mathematical creation and to additive manufacturing may be based on a voxel description of such structures. For instance, it dramatically simplifies the combination and merging of different structures, features (holes and stabilizer rods) and elements (hulls and screens) by Boolean operations and is naturally compatible with multimaterial object definition. A simple, straightforward export interface to 3D printing devices is possible via xy-image stacks, instead of error-prone *.stl topology files. A few open-source and commercial voxel based geometry generators are available (VoxelBuilder, MKagicaVoxel, GeoDict, Monolith, etc.), which are mainly oriented towards game development and are often sparsely prepared for mathematical based structure generation. Alternatively, advanced, state-of-the-art mathematics software platforms (Mathematica, Matlab) include all the necessary tools (mathematical function/large matrix/logical operation + visualization + parallelization) to fulfill the requirements. An example for this approach is presented in Section 3.1.1.

In addition, voxel models can describe a surface in higher discretization accuracy by the extension to “Distance Fields”. Instead of providing a voxel with the information whether it lies inside or outside an object, the Euclidean distance to the surface is stored. Furthermore, it is determined whether the voxel coordinate is inside the object or not. If it lies inside, the distance is taken as a negative value. A so-called signed distance field is created [19,20].

The Boolean operations described above are determined by voxel-wise comparing two voxel datasets. It applies to the union of two objects, dist(A∪B)=max(dist(A),dist(B)), for the difference dist(A−B)=min(dist(A),−dist(B)) and for the intersection dist(A∩B)=min(dist(A),dist(B)). Another advantage of these distance fields is the calculation of offsets. Only the scalar voxel intensities are calculated by the given distance. A conversion to isosurface models (e.g., triangle meshes) is possible without problems with known methods like the Marching Cubes algorithm [21]. The advantage over the binary model lies in the more detailed representation of an object within a voxel data structure. Figure 1 illustrates this advantage.

### 2.2. CerAM Replica of Single-Material Porous Ceramics

One advantage of the replica technique is the opportunity to combine a well-established technology with the use of innovative, 3D-structured components as templates. Designed for special purposes, the structural properties of that kind of replicated porous ceramics, such as pressure drop and geometrical surface area, correspond directly to the previously well-defined parameters.

Another important aspect is the hollow space left by the decomposed polymeric template after debinding and sintering the ceramic components. While the hollow struts of ceramic foams are typically not useable but crucial with respect to the mechanical stability (sharp edges of the triangular polymeric foam strut), the cavities of porous ceramics based on additively manufactured, polymeric templates can be functionalized. Like buried channels in the walls of the structure, the cavities might be used for internal cooling or as an intensification of the heat exchange in a reactor (heating/cooling). Additionally, by using additive manufactured templates, it is possible to feed a reactor via perforations in the wall and their connection to the template cavities, which act as internal pipeline.

The CerAM Replica represents the combination of the replica technique with 3D-structured polymeric templates that have been manufactured by AM. In this process, the polymeric 3D structures are coated with a ceramic suspension similar to reticulated foams or technical textiles. The suspension includes ceramic particles, organic and inorganic additives, such as dispersing agents, temporary binder, rheological additives, and sintering aids. For an environmentally friendly preparation and to avoid expensive equipment for explosion prevention, the suspensions are usually water-based. The impregnation of the template is comparable to a dip coating. Excess suspension blocking the aspired open space in the structure can be removed by using rollers for flexible templates, or a centrifuge for rigid ones. This homogenization step is also useful to adjust a well-defined coating layer on the polymeric template. For both, the primary coating of a template and a subsequent coating of thereby created complex ceramic structures with additional layers, such as catalytic active washcoats, the exact adjustment of the suspension’s rheology is mandatory to create a uniform coating. In that matter, the slurry should show a shear-thinning behavior in order to easily impregnate the struts and walls as well as remove the excess material, to achieve an even coating. Simultaneously, similar to wall paint, the slurry shall be characterized by a well-defined yield point to avoid any draining effects [22,23,24]. The adjustment of the yield point depends on the geometry and the aspired coating layer thickness. By varying the solid content and using one or more rheological additives, such as polyurethane thickener: a homogeneous coating is adjustable. 

By applying different coating suspensions and partially covering segments of the structure, connections between the later template-related cavities and the usable space can be achieved. Subsequently, the coated templates are dried, followed by the thermal treatment including the decomposition of the polymeric template and the sintering of the ceramic material.

### 2.3. CerAMfacturing of Multiproperty Porous Ceramics 

Different challenges exist concerning the AM of multimaterial or multiproperty ceramics. Two or more different materials or suspension compositions have to be processed simultaneously in one AM process. Direct working AM technologies [25] are more suited than indirect working AM technologies because of the selective deposition of the material instead of the selective curing of a material, which was deposited on the entire building area. Latter requires a removing process for the first material to make room for the deposition of the second material or feedstock.

CerAM T3DP (Thermoplastic 3D Printing) is an AM technology, which is based on the selective deposition of molten, highly particle-filled thermoplastic suspensions as single droplets [26,27]. These droplets are fused to generate 3D structures before the suspensions solidify because of the increasing viscosity resulting from cooling and the absence of shear forces. The portfolio of processible materials is nearly unlimited, which could be demonstrated for alumina, zirconia [26], and cemented carbide [28]. Furthermore, the suitability for AM of ceramic-based multimaterial and multiproperty components could be demonstrated, e.g., for combinations of stainless steel and zirconia [29,30], white, and black zirconia [31], as well as the combination of dense and porous zirconia in one AM component [32].

The last combination is very interesting for applications such as implants or catalytic support structures. But for the latter application alumina will be more interesting than zirconia because of the higher thermal conductivity and the lower material costs. The AM of alumina components by CerAM T3DP, which combines dense and porous volumes inside, is one topic of this paper and will be demonstrated below.

## 3. Materials and Methods 

### 3.1. CerAMfacturing of Porous Ceramics by CerAM VPP

#### 3.1.1. Generation of CAD Data by Voxel Based Geometry Generators

A voxel based approach was applied to design and manufacture a demonstrator prototype for a novel three domain chemical reactor structure with integrated heat exchanger/cooler structure. The approach is based on a space-filling interwoven 3D network of three domains which are separated from each other.

Two domains (fluids B and C), separated by perforated walls, may contain reactants, while the third domain (fluid A), enclosed by impermeable walls, may contain a heat carrier (or coolant) fluid (Figure 2). An integrated housing and media supply structure was formed as well to separate the main flow directions of the three fluids along the x, y, and z directions by masking the other domain areas at the housing inlet areas with shielding walls.

Mathematical basis of the interwoven three domain structure is the gyroid function:$$K=\cos\left(\frac{2\pi }{\Delta L_{x}}\cdot \left(x-x_{0}^{}\right)\right)\cdot \sin\left(\frac{2\pi }{\Delta L_{y}}\cdot \left(y-y_{0}^{}\right)\right)+\cos\left(\frac{2\pi }{\Delta L_{y}}\cdot \left(y-y_{0}^{}\right)\right)\cdot \sin\left(\frac{2\pi }{\Delta L_{z}}\cdot \left(z-z_{0}^{}\right)\right)+\cos\left(\frac{2\pi }{\Delta L_{z}}\cdot \left(z-z_{0}^{}\right)\right)\cdot \sin\left(\frac{2\pi }{\Delta L_{x}}\cdot \left(x-x_{0}^{}\right)\right)$$
where choice of the isosurface parameter *K* controls the position of the domain-wall surfaces; while $\Delta L_{x}$, $\Delta L_{y}$, $\Delta L_{z}$ define the period length of the structure along the x, y, and z directions (may be functions of space to generate graded structures), respectively, and parameters $x_{0}$, $y_{0}$, and $z_{0}$ allow a shifting of the structure in space. For the current example the fluid domains were specified by
fluid A: K≤−0.95;fluid B: 0.95≤K;fluid C: −0.65≤K≤0.65 andperiod length of the gyroid selected as $\Delta L_{x}=\Delta L_{y}=\Delta L_{z}=6\;\mathrm{mm}$.

The 3D gyroid core structure is shown in Figure 3. The voxel length was initially defined to 80 µm in x- and y-direction and 50 µm in z-direction. Later it was reduced to 40 µm and 25 µm (corresponding to the physical resolution of the CerAM VPP device) by resizing the voxelized component structure by an array resample with scale factor 2 via linear interpolation. The reason was to reduce the computing time and storage challenges from large 3D array operations. The total dimension of the component (including integrated inlet and outlet connects) was 1920 × 1080 × 1200 voxel.

The geometry definition was implemented in Mathematica 11, taking advantage of the comprehensive toolset of integrated functions and procedures for array and logical operations, optimized for the use of large arrays by condensed storage and parallelization. The developed workflow may be easily applied as a template to generate other structures and components.

Additionally to the generation of the gyroid structure, a set of various parameterized cylinder and cuboid domains was created as basis domains for the cylindrical housing wall and pipe supply structures, shown in Figure 4. The full component geometry was shaped by Boolean combination of the various basic domains in voxelized form. The perforation in the wall between fluid A and B was created by applying a pattern of cylindrical holes to that wall structure, masking areas where the angle between drill and wall would become too steep. All domain and combining operations could be easily formulated by means of mathematical functions and operators.

For the generation of the structure a 256 GByte, a 32-thread Windows workstation was used, the total computation time for the full structure was about 10 h. The 3D voxel array was exported as an image stack of 1200 (*.png) 1920 × 1080 individual images to the software belonging to the CeraFab 7500 device.

#### 3.1.2. Generation of CAD Data based on Distance Fields

The generation of porous structures based on Distance Fields is done by specifying a grid topology. This means a network of beam and node elements. A local dimension can be added to these. This allows the realization of gradations within the porous structures (see Figure 5). Simple lattice topologies can be regular beam structures; however, complex, irregular topologies are also possible. Figure 6 shows as an example of foam-like structures, which were created with a topology based on voronoi diagrams. The design space of a grid or porous structure is described by a signed distance field of the object. This allows to create the structure on the one hand, and to mask later solid areas within the object on the other hand. A signed distance field is also calculated for each topology element. Basic elements such as cylinders, cones, and spheres can be used, but complex geometries are also possible, e.g., for adapting individual cross-section geometries. Boolean operations follow (see Section 2.1). A distinction is made between whether the generation is subtractive or additive. With the subtractive method, the geometry of the topology is subtracted from the design space model: dist(R)=dist(A∪!B). Using the additive method the geometry of the topology is generated directly: dist(R)=dist(A∪B). The subtractive method allows a setup of defined pore sizes, which may be necessary for medical applications (Figure 7).

#### 3.1.3. CerAM VPP

The CerAM VPP process is an AM process for ceramic components which is used at Fraunhofer IKTS and is based on the digital light processing (DLP) technology. This technology was commercialized by Lithoz GmbH (Vienna, Austria), which sells suspensions and AM devices under the trade name “lithography-based ceramic manufacturing (LCM)” [15]. The AM device CeraFab7500 of Lithoz was the first commercially available printer for AM of dense ceramic components with a lateral resolution of 40 µm (pixel-size; 635 dpi) in x–y direction and 5–100 µm (layer thickness) in z-direction [11]. Complementary to DLP, a photoreactive suspension was deposited as complete layer selectively exposed to photons with a wavelength of 465 nm and a maximum intensity of 32.7 mW/cm². In various articles detailed descriptions of the process have been published [15,16,17,18,33,34]. As compared to other AM technologies for ceramic components, the main advantages of this technology are the high resolution, the high density of the sintered components as well as the surface quality of the final components. 

Within this study a commercially available alumina suspension (Lithalox 350D; Lithoz, Vienna, Austria) was used to build the specified ceramic demonstration components by CerAM VPP. The cleaning of the green bodies as well as the debinding and sintering (1650 °C, 2 h) were done according to the instructions given by Lithoz.

### 3.2. CerAMfacturing of Single-Material Porous Ceramics by CerAM Replica

As demonstration example for CerAM Replica, a complex, axially symmetric structure made of pressureless sintered silicon carbide was prepared using a polymeric template generated by stereo lithography. In this demonstrator component cross-linked and tapered struts are circumferentially arranged around a perforated cylinder. The structure was designed by means of SolidWorks and transferred to a real component using the CeraFab 7500 system from Lithoz. As template material a mixture of different monomers, oligomers and photoinitiators, which is normally used as binder system for CerAM VPP suspensions, was chosen. 

The ceramic coating suspension consisted of a SiC powder with a bi-modal particle size distribution and a highly carbon containing temporary binder. As sintering aids boron and carbon were used, added in the typical quantity of 0.5 and 2.5%. After the impregnation the samples were centrifuged for 10 s at 200 rpm. After the drying at 80 °C the samples were pyrolyzed at 1200 °C in order to decompose the template. In order to avoid crack formation in the green body during that step, a heating rate of 0.5 K/min is advantageous in the temperature range between 180 and 400 °C where the main decomposition of the polymeric template occurs. The heat treatment was finalized with a sintering step at more than 2000 °C for ~1 h under inert atmosphere. The final reactor component of pressureless sintered silicon carbide (SSiC) was intensively characterized by computer tomography (CT) analysis (see below). 

### 3.3. CerAMfacturing of Multi-Properties Porous Ceramics by CerAM T3DP

Figure 8 shows the CAD model of a honeycomb consisting of two different parts. Each part will be printed with a different suspension to achieve dense (red) and porous (green) volumes. The CAD data were designed by means of SolidWorks. To generate the machine data the model was transformed into a g-code using the Slic3r software (version 1.2.9) as shown in Figure 8b.

To prepare multifunctional components, like dense-porous components, particle-filled thermoplastic suspensions are needed leading to dense and porous volumes after debinding and sintering. However, the dispensing of the suspension has to be guaranteed. Three alumina powders with different characteristics were used. Table 1 shows the used powders with their specifications.

For each powder thermoplastic suspensions with different contents of pore forming agents (PFA) were prepared. Starch was used as PFA. The compositions of the prepared suspensions are presented in Table 2. The binder system, which consists of a mixture of different waxes, was molten at 100 °C and stirred afterwards for about 15 min. Then a dispersant and the PFA were added and stirred for further 15 min. Afterwards, the powder was added and homogenized for 2 h.

To investigate the density, which can be achieved with the different suspensions, small samples (16 mm × 16 mm × 10 mm) were printed with each suspension and sintered at 1250 °C or 1600 °C for 2 h. The density of the samples was measured in accordance with DIN EN 623-2. 

Different pairs of thermoplastic suspensions were used to manufacture the multimaterial components by CerAM T3DP. The red marked areas in Figure 8 were printed with a suspension without any additives, which results in dense microstructure in the sintered component. This is necessary to achieve a sufficient strength and gas impermeability. To realize porous structures with high specific surface area (marked green in Figure 8) suspensions with PFA were used. Tests with suspensions with a PFA content of more than 10 vol.% were not successfully, because it was not possible to reproducibly deposit the suspension.

## 4. Results

### 4.1. CerAM VPP 

#### 4.1.1. CerAM VPP—Machine Files

The final designs were sliced by special slicing tools to generate the exposure maps for each layer (1-bit black-and-white image), which could be transferred directly to the software of CeraFab 7500 (CeraFab-DP Version 7.65.7.9).

#### 4.1.2. CerAM VPP—Sintered Components

Figure 9 shows the sintered alumina components manufactured by CerAM VPP. On the left side the sliced version of the reactor component is shown. It was sliced within the CAD data and manufactured as ¾-component to make the inner structure visible. All channels and even the small holes between the different systems are open. On the right side an alumina component according to Figure 6b with graded porosity and pore size distribution is shown.

#### 4.1.3. CerAM VPP—Characterization

One of the most challenging steps within the process chain for CerAM VPP is the cleaning step. Any noncured suspension has to be removed to avoid the clocking of the inner channels. CT scans were done to assess the inner structure of the sintered components. Figure 10 shows two images of cross-sections for the complete reactor component, oriented in x-y- and x-z-directions. It becomes visible that all channels and holes are open. Only at the bottom residues of noncured suspension can be detected, but in this section of the component it is not critical. The accumulation of noncured suspension is a typical phenomenon for CerAM VPP, because during thermal treatment the viscosity and wetting behavior are decreased. Due to this, the noncured suspension accumulates at the bottom of the component because of gravity, before it is cured thermally, debinded, and sintered.

Figure 11 shows different images of the alumina component with graded porosity (Figure 9b). On the left side the fine pore structure at the bottom of the component is shown, on the right side the structure with bigger pores on the top. In between there is a cross-section in the x-z-direction generated from the CT data and making the vertically graded porosity visible.

### 4.2. CerAM Replica

#### 4.2.1. CerAM Replica—Final CAD Data and Polymeric Template

The prepared CAD model of the structure was converted into a compatible format for stereo lithography (*.stl). A perforated cylinder with a wall thickness of 1.1 mm, surrounded by tapered struts with a minimal diameter of 0.3 mm, was designed. In Figure 12 the polymeric template, additively manufactured using the Lithoz device, is depicted.

#### 4.2.2. CerAM Replica—Green and Sintered Components

In case of the SiC suspension, a homogeneous coating of all struts was found while all the defined pores between the strut network remained perfectly open. During pyrolysis, the melting and outgassing template material induced small cracks at the base area, especially in the center ring holding the highest amount of template material. In contrast to this, all the other struts have survived the pyrolysis without any defects (Figure 13a). Despite shrinkage, no new cracks were observed in the structure after completed sintering (Figure 13b).

#### 4.2.3. CerAM Replica—Characterization

The CT analysis of the sintered component offers an interesting insight into the cavities created by the decomposed template and indicates the homogeneous coating of all struts (Figure 14).

It is clearly visible that the buried channels of various diameter and connectivity were realized. The sintered SSiC leads to a mostly dense wall, separating the cavities from the surrounding open space in the structure. The accurately tapered tips at the end of the struts could be prepared as small jet nozzles, e.g., for burner structures. 

### 4.3. CerAM T3DP 

#### 4.3.1. CerAM T3DP—Characterization of the Suspensions

Table 3 summarizes the calculated open porosity as result of the density measurements on the sintered test samples. For all three suspensions (A, D, and H) the remaining open porosity after sintering at 1600 °C was below 0.5%; also, the remaining open porosity of the samples manufactured with the suspensions with PFA was very low after sintering at 1600 °C.

The maximum content on PFA in the suspensions was limited to 10 vol.%, because suspensions with higher amounts of PFA could not be reproducibly processed. But this porosity is too low for the addressed application. That is why the residual open porosities were investigated after sintering at 1250 °C. Due to the different sintering activity of the used powders the differences in the residual open porosity was much higher than after sintering at 1600 °C. For suspension D (based on SMA6), a denser microstructure resulted, because of the higher sintering activity of the fine powder, whereas the powders CT1200 SG and MR52 were sintered only partially.

To increase the difference in the residual open porosities for the two suspensions B (CT1200 SG + PFA, open porosity: 30.22%) and D (SMA6, open porosity: 20.64%) the solid content of the suspensions were adjusted more precisely. The solid content of suspension D was further increased as long as the printability was given, whereas the solid content of suspension B was decreased as long as sedimentation occurred. A big challenge was the different shrinkage behavior of the suspensions. To process two suspensions together, a nearly equal shrinkage is necessary to avoid cracks. These developments resulted in the suspensions C and G, which was the best combination to produce nearly defect-free sintered multiproperty components according to the CAD model in Section 3.3.

#### 4.3.2. CerAM T3DP—Green and Sintered Components

Different combinations of suspensions with and without PFA were used to manufacture multiproperty components by CerAM T3DP. Figure 15 shows some green components, manufactured with two different suspensions. However, nearly all manufactured components contained small cracks after sintering at 1250 °C (Figure 16). Only some of the components based on suspension C and G could be sintered without defects (Figure 16c).

To avoid cracking, the CAD model was adapted and an additional cylindrical tube was added at the outside of the honeycomb. This structure was manufactured with the suspension C. This porous area had no technological advantage but the shrinkage of the suspension G, which is higher than for suspension C, was limited. This further increased the component’s quality after sintering.

#### 4.3.3. CerAM T3DP—Characterization

To characterize the sintered CerAM T3DP components, SEM images of cross-sections were taken. In Figure 17 the microstructure at the interface between the dense and the porous volumes is shown in different magnifications. A very good connection between the two volumes is visible as well as the different microstructures, resulting from the different powders and the added PFA. 

## 5. Conclusions

Different strategies for manufacturing of porous ceramic components were demonstrated. All these strategies are based on AM technologies and create new opportunities concerning the design and composition of the porous components, but there are specific restrictions as well. 

Because of the high resolution and the very good properties of the sintered ceramic components, CerAM VPP is a very interesting approach for AM of ceramic components with a macroscopic porosity. Furthermore, it would be possible to increase the porosity within the ceramic structures by the addition of PFA or the targeted variation of the sintering temperature. One of the main challenges is to generate the required CAD data. Two different approaches were presented, which allow the manufacturing of ceramic components with geometries, which have not been realized so far. To achieve high accuracy, a further challenge is to clean the small pores that are deeply located within a structure. But in case of success, the CerAM VPP technology will be one of the most interesting AM methods to realize designed lattice pore structure of high resolution.

CerAM Replica is a promising approach to generate innovative, porous ceramics. Based on a well-known process, this method enlarges the range of applications of porous ceramics due to its structural and functional flexibility. Especially the exploitation of the process-related cavities for the transport of media and/or reactants is remarkable. In addition, this method helps to overcome the lack of AM of special ceramics such as pressureless sintered silicon carbide.

As a direct working AM technology CerAM T3DP is suitable for AM of multimaterial and multiproperty components like dense porous alumina components as demonstrated in this work. Further investigations are needed to identify optimal suspension compositions and process parameters for additive manufacturing nearly defect-free sintered, dense porous ceramic components and for increasing the porosity of the porous area. 

## Figures and Tables

**Figure 1 materials-12-00663-f001:**
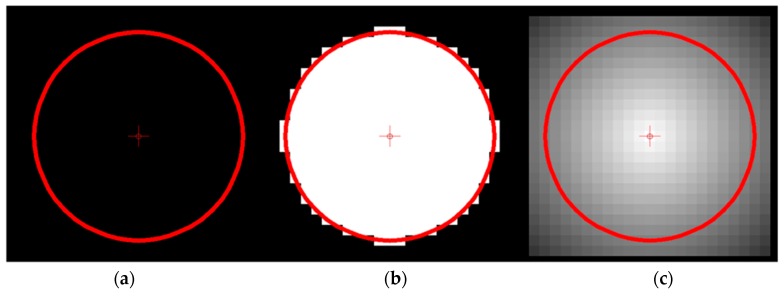
The approximation of a given circular contour (**a**) in a binary voxel model (**b**) allows only an inaccurate description. The use of signed distance fields (**c**) and the interpolation-based recalculation of the contour is much more accurate.

**Figure 2 materials-12-00663-f002:**
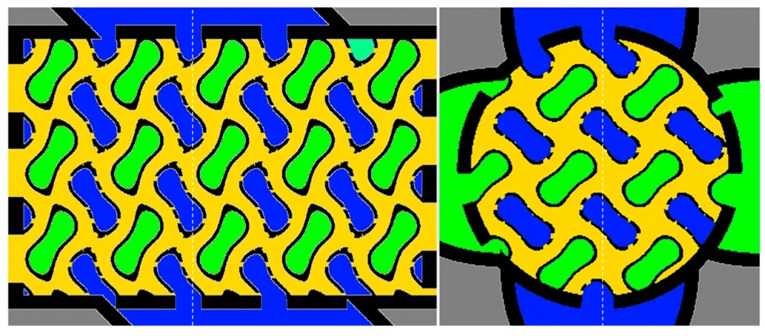
2D sections of the reactor component to visualize the interconnected domain structure of fluid domains A (green), B (blue), and C (yellow). White-dashed lines label the position of the sections.

**Figure 3 materials-12-00663-f003:**
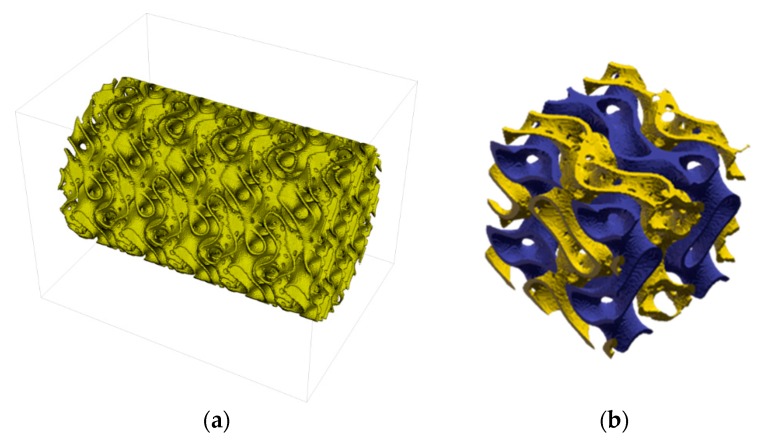
3D gyroid core structure of the reactor component ((**a**) full domain; (**b**) detail; blue walls separating fluid A and C and yellow walls, separating fluid B and C).

**Figure 4 materials-12-00663-f004:**
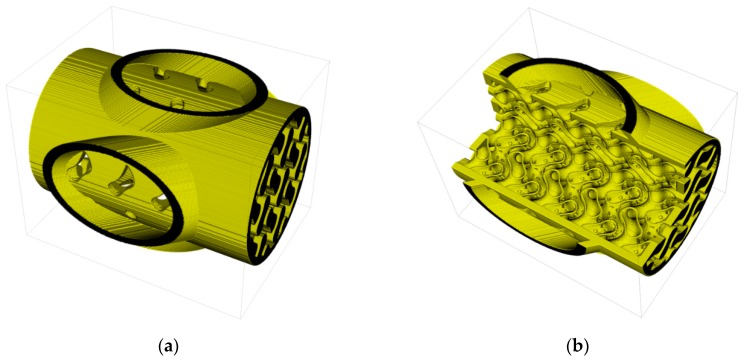
Housing of the gyroid structure (**a**) and sliced component (**b**).

**Figure 5 materials-12-00663-f005:**
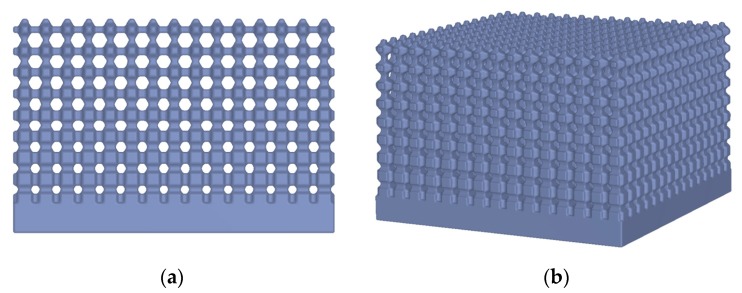
Graded porous structure of a cuboid (**a**) side view; (**b**) perspective view. The pore size increases with increasing component height.

**Figure 6 materials-12-00663-f006:**
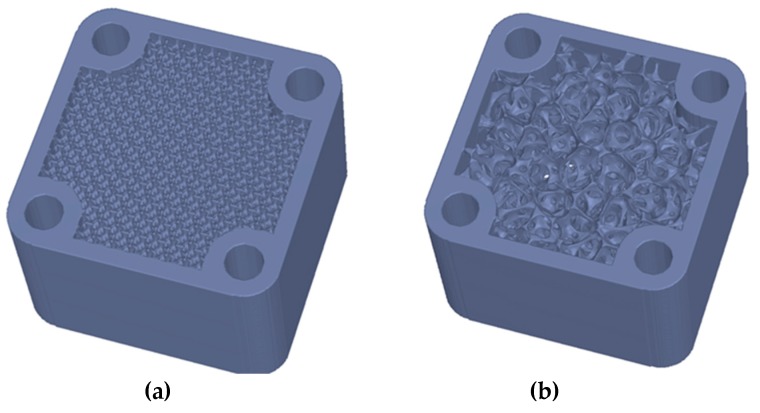
Examples of porous structures. (**a**) A regular structure within the component geometry. (**b**) A foam-like structure based on voronoi diagrams is depicted.

**Figure 7 materials-12-00663-f007:**
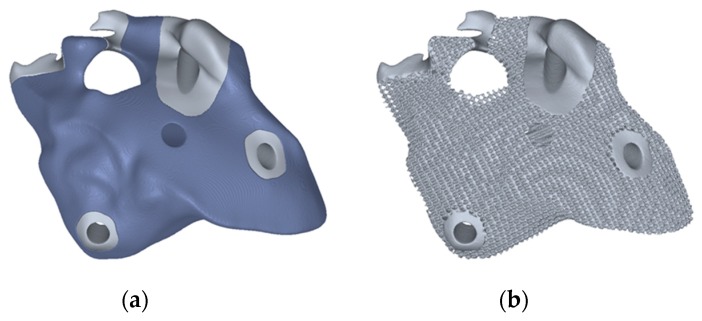
Another example shows the use of porous structures within an implant geometry. (**a**) The definition of the design space for the lattice structure (dark blue); (**b**) The final implant design.

**Figure 8 materials-12-00663-f008:**
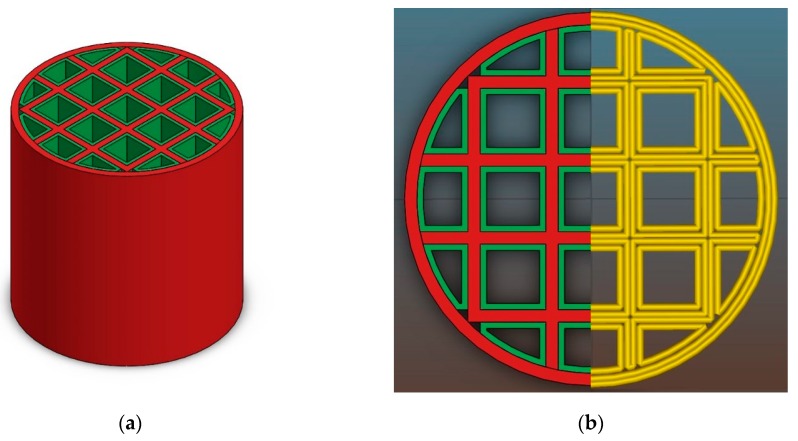
CAD model of a honeycomb consisting of two different materials (**a**) and visualization of the machine data (**b**).

**Figure 9 materials-12-00663-f009:**
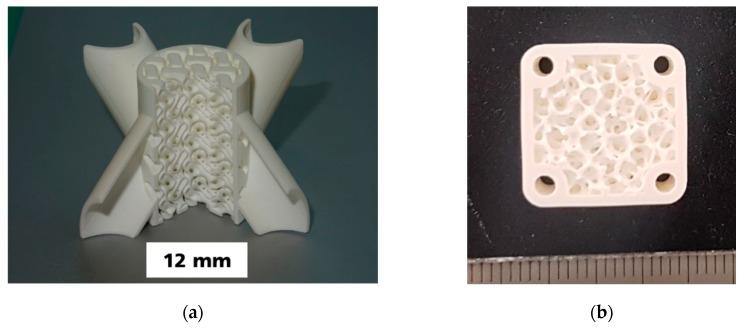
Sintered alumina components manufactured by CerAM VPP: Sliced version of the reactor component (**a**), ceramic component with graded porosity, and pore size distribution inside (**b**).

**Figure 10 materials-12-00663-f010:**
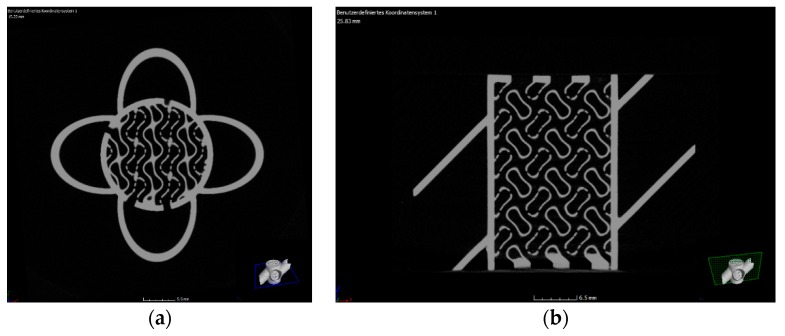
Cross-section images recalculated from the computed tomography (CT) scan of the sintered alumina reactor component (outer diameter of reactor: 12 mm): horizontal cross-section (**a**), vertical cross-section (**b**).

**Figure 11 materials-12-00663-f011:**
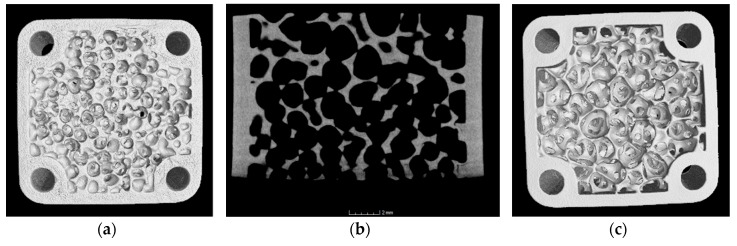
CT scan images of the sintered alumina component (18 × 18 × 10 mm³) with graded porosity: bottom (**a**), top (**c**), and cross-section image recalculated from the CT scan (**b**).

**Figure 12 materials-12-00663-f012:**
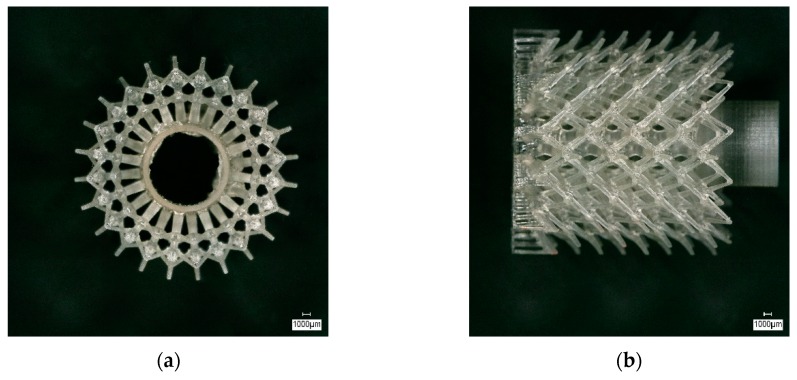
Complex structured polymeric template: top view (**a**) and side view of generated polymeric template by stereo lithography (**b**).

**Figure 13 materials-12-00663-f013:**
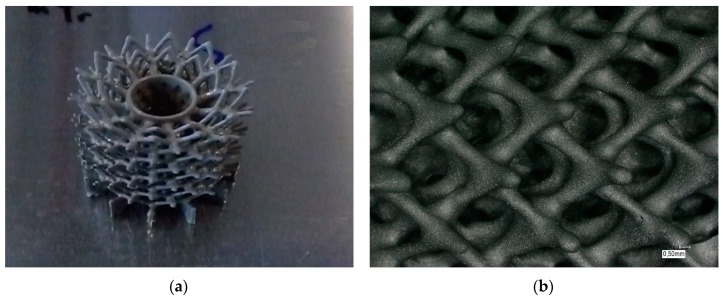
Replication of the polymeric template: polymeric template with dried ceramic coating before heat treatment (**a**) inner diameter of hole: 8 mm and (**b**) sintered SSiC struts.

**Figure 14 materials-12-00663-f014:**
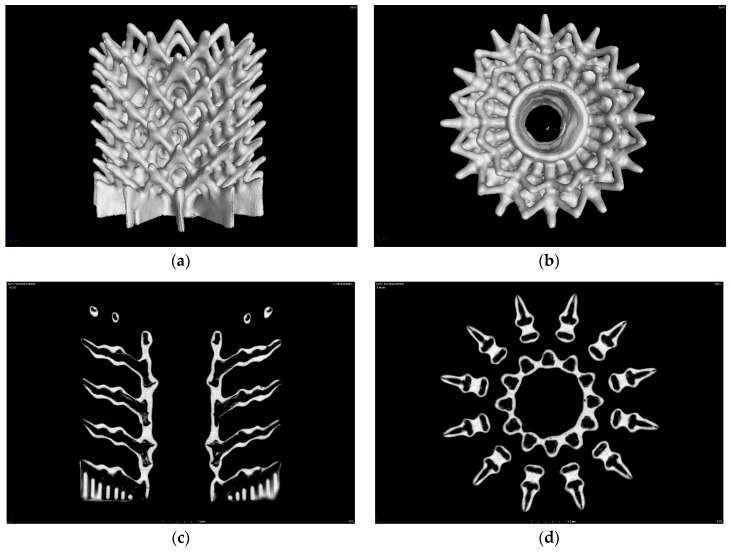
CT analysis of the sintered silicon carbide (SSiC) structure (outer diameter approximately 25 mm): side view (**a**), top view (**b**), vertical cut (**c**), and horizontal cut (**d**).

**Figure 15 materials-12-00663-f015:**
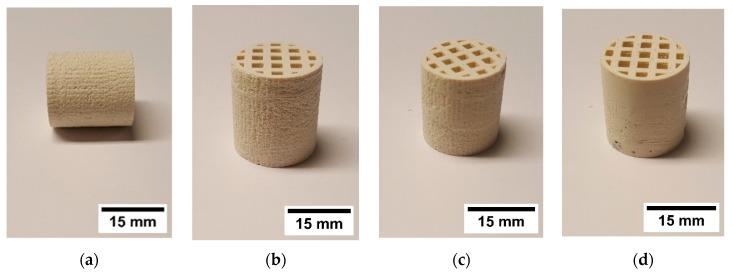
Green honeycombs additively manufactured with different suspensions: (**a**) C&G, (**b**) C&G, (**c**) C&G, and (**d**) A&F.

**Figure 16 materials-12-00663-f016:**
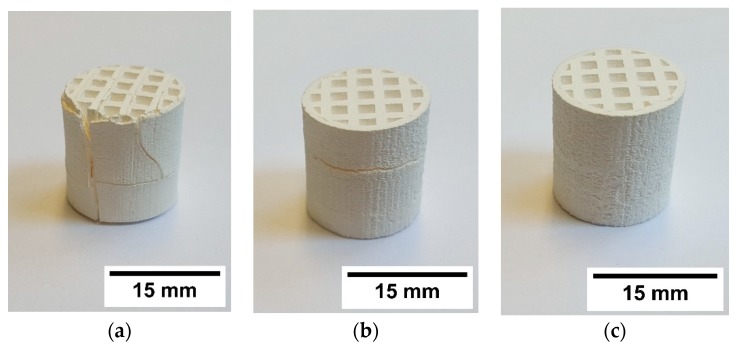
Sintered honeycombs manufactured with different suspensions: (**a**) A&F, (**b**) C&G, and (**c**) C&G.

**Figure 17 materials-12-00663-f017:**
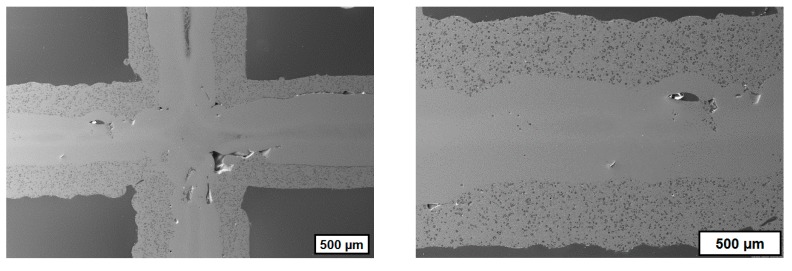
SEM images in different magnifications of cross-sections of sintered ceramic dense porous components manufactured by CerAM T3DP.

**Table 1 materials-12-00663-t001:** Characteristics of the used alumina powders.

Name	Company	d_50_ in µm	d_90_ in µm	Fired Density in g/cm³
CT 1200 SG	Almatis	1.734	3.14	3.99
Baikalox SMA6	Baikowski	0.251	0.671	3.96
MR52	Martinswerk	1.836	4.856	-

**Table 2 materials-12-00663-t002:** Compositions of the prepared suspensions.

Component	Content in vol.%								
	A	B	C	D	E	F	G	H	I
CT1200 SG	50	50	45						
SMA6				35	35	45	55		
MR52								50	50
binder system	45	35	40	60	50	50	40	45	35
dispersant	5	5	5	5	5	5	5	5	5
PFA	0	10	10	0	10	0	0	0	10

**Table 3 materials-12-00663-t003:** Calculated open porosity as a result of the density measurements.

Suspension	CT1200 SG	SMA6	MR52	PFA (10 vol.%)	Open Porosity [%] (After Sintering at 1250 °C)	Open Porosity [%] (After Sintering at 1600 °C)
A	x				28.85	0.46
B	x			x	30.22	4.45
C	x			x	24.05	
D		x			20.64	0.18
E		x		x	24.44	2.93
F		x			16.83	
G		x			8.49	
H			x		23.03	0.09
I			x	x	25.25	0.51

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
