# Peer review of "Alternative Process Routes to Manufacture Porous Ceramics—Opportunities and Challenges"

_materials, 2019, doi:10.3390/ma12040663_

Reviewer 1 Report

This paper discusses the processing of porous ceramics using various additive manufacturing methods. It is a good paper after minor revision and my comments are as below:

Figures 10, 11 and 14 do not have scale bars. Authors may want to add scale bars for these figures.

Page 12, line 377, authors mention the suspension's rheology. Authors may want to provide more characterizations of the SiC suspension's rheology behaviors.

Author Response

Dear reviewer,

thank you very much for your comments, which helped us to improve our manuscript. We have added the missing scale bars. Unfortunately, we could not provide more characterizations of the SiC suspension's rheology behaviors, but this will be one topic of our future works.

Reviewer 2 Report

Thanks to authors for interesting and actual research

All paper parts well completed and paper in common is cohesive.

However, I suggest to authors enhance the introduction, to highlight the actuality of research.

Author Response

Dear reviewer,

thank you very much for your comments, which helped us to improve our manuscript. We have added some more references and the number of papers concerning porous ceramics founded at scopus to highlight the actuality of research.

Reviewer 3 Report

The manuscript “Alternative process routes to manufacture porous ceramics – Opportunities and Challenges” deals with the limitations of current CAD software regarding additive manufacturing by 3d printing. In this work, the authors provide an alternative approach using a different mathematical description to generate the CAD models to print, and they apply this method in 3 different ways to generate several ceramic parts with various complex geometries and densities. I believe the manuscript is of sufficient interest and quality for publication in this Journal, although some minor revisions are needed.

1.       In general, a revision of the English language is needed. Some words are misused (For example the word “realized” in line 32, or in the title where instead of “process routes”, it should be “processing routes”).

2.       There is a lack of uniformity in the writing style throughout the manuscript. The authors should have external people proof read the manuscript and try to homogenize the style.

3.       Certain claims made in the introduction are not entirely accurate:  In line 36, the authors claim that by slightly sintering ceramic particles, porosities up to 40% can be obtained. There are several papers published that show higher porosities obtained with this method, for example “Caccia, et Al. Ceramic–metal composites for heat exchangers in concentrated solar power plants, Nature 562, 2018” where WC powder is sintered in bodies with 53% porosity. Similar claims are made for other methods to prepare porous ceramics. More references are needed when providing this maximum attainable values. I recommend the authors examining the work of F. Ravera and L. Liggieri.

4.       In general, a more diverse and extensive literature review would be beneficial for this paper. The number of autocitations is rather high. Giving that additive manufacturing of ceramics is quite a “hot topic” these days, authors should try to include more citations of other research groups as well.

5.       Since the language of the Journal is English, the use of citations to papers written in other languages should be avoided unless it is a classic paper.

6.       In line 123, the use of the word “chapter” doesn’t seem like the most appropriate one, I recommend using the word “section” instead.

7.       The use of excessive subdivision (i.e. section 3.2.2) makes the paper hard to read. The authors should use only one subdivision and combine the different sections for a more fluent reading.

8.       The main goal of this work is loosely stated between the introduction and the Materials section. I would recommend the authors to have one, succinct, clear paragraph stating the main goals of the work.

9.       In section 3.3.2 the authors provide a characterization of the alumina particles used in this work (D50, D90 and sintered density). It is not clear from the text where these values come from. If the particle size distribution was provided by the vendors, please clearly state so. If it was measured by the authors please include a description of the measurement and move table 1 to the Results section. In Table 1, the fired density of MR52 is missing.

Author Response

Dear reviewer,

thank you very much for your comments, which helped us to improve our manuscript significantly. We have added some more information concerning your comments and more references and we revised the English grammar and style.

Reviewer 4 Report

This paper describes three processing routes of porous ceramics using the AM approach. The developed routes are suitable for materials with different architectures and compositions. The quality of the illustrative material in the paper is very good, the presentation is logical. I think this paper requires only minor revision before publication.

This article has features of a focused review. In my opinion, more literature citations should be added to publications describing more traditional methods and other advanced methods of manufacturing of porous ceramics.

It seems that, as this article is a focused review, it does not have to be divided into the same sections as a regular article (Materials and Methods, Results).

Please describe in more detail the debinding and sintering processes and problems associated with them (selection of the treatment temperatures and durations).

There are some typos and missing commas in the text. 

Author Response

(The authors gave the same response as above.)
